# ON THE DYNAMICS OF LEARNING
# TIME-AWARE BEHAVIOR WITH RNNS

## ABSTRACT

Recurrent Neural Networks (RNNs) have shown great success in modeling time-dependent patterns, but there is limited research on how they develop representations of temporal features during training. To address this gap, we use timed automata (TA) to introduce a family of supervised learning tasks modeling behavior dependent on hidden temporal variables whose complexity is directly controllable. Building upon past studies from the perspective of dynamical systems theory, we train RNNs to emulate a new class of TA called temporal flipflops, and we find they undergo *phase transitions during training* characterized by sudden and rapid discovery of the hidden time-dependent features. In the case of periodic "time-of-day" aware flipflop, we show that the RNNs learn stable periodic cycles that encode time modulo the period of the transition rules. We then use fixed point stability analysis to monitor changes in the RNN dynamics during training, and we observe that the phase transition coincides with a *bifurcation* from which stable periodic behavior emerges. We also show that these cycles initially lose stability if the RNN is later trained on the same TA task but with a different period, and we explain this result through analysis of a simple differential equation for learning oscillations via gradient flow. Through this work, we demonstrate how dynamical systems theory can provide insights into not only learned representations, but also the dynamics and pathologies of the learning process itself.

## 1 INTRODUCTION

Recurrent neural networks (RNNs) (Elman, 1990), long-short term-memory networks (Hochreiter & Schmidhuber, 1997), and gated recurrent networks (Chung et al., 2014) are some of the most widely used machine learning models for learning temporal relationships. Their ability to store and manipulate external inputs over time have made them popular in sequence related tasks such as time series prediction, language translation, or control.

Despite time-dependence being central to recurrent architectures, past literature has placed little emphasis on how artificial RNNs utilize time itself in their computations (Bi & Zhou, 2020) and how their temporal representations *develop* during training. The ability to deduce temporal patterns is a fundamental skill required of artificial and biological agents; for instance, humans decompose time into smaller repeating blocks (days, weeks, etc.) instead of viewing each year as a sequence of distinct days. Architectures that learn such modular representations of time will allow for smaller models with increased generalization.

We ask if recurrent networks learn to be similarly temporally aware, and if so, how do they discover hidden temporal structure. We approach these questions from the perspective of mechanistic interpretability, specifically *developmental interpretability* (Hoogland et al., 2023) — the practice of analyzing how deep learning models change through training as a means of understanding the representations they ultimately learn. Through this approach, researchers have been successful in explaining training phenomena such as grokking (Liu et al., 2022; Chughtai et al., 2023) and in-context learning in transformers (Olsson et al., 2022). Overall, this interpretability research aims to improve the reliability and safety of deep learning models.

Our works performs a similar developmental analysis in the context of time-aware recurrent models. To enable this study, we use timed automata (TA) (Alur & Dill, 1994) to introduce a new family of time-aware sequence processing tasks that give researchers direct control over the complexity

of the time-awareness needed to solve them. These tasks are designed to allow scalability and customization which enable the testing of different forms of time-awareness. Our work draws upon the long history of defining computational capabilities through automata, and it extends past research on training neural networks to emulate automata behavior (Pollack, 1991; Tino et al., 1998; Zeng et al., 1993; Arai & Nakano, 2000; Michalenko et al., 2019; Oliva & Lago-Fernández, 2021; Dan et al., 2022) to include the time-dependent behavior described by timed automata.

Our analysis also extends existing research using dynamical systems theory to analyze RNNs trained to emulate automata. Past studies on time-*in*dependent automata found that dynamics about stable fixed points encode the automata states, and input symbols induced state transitions by switching the networks' states between basins of attractions. Sussillo & Barak (2013) discovered heteroclinic orbits connecting these fixed points, and Ashwin & Postlethwaite (2020) used this behavior to construct continuous-time RNNs with behavior like finite-state machines. To our knowledge, RNNs have never before been trained to emulate the behavior of time-dependent automata.

We analyze the dynamics of recurrent networks both *during* and *after* learning to shed light on the development of their learned representations. Post training we find that the networks learn reusable behaviors of time that significantly improve learning and generalization, with the entire sequence quantized into the smallest time period required to express the rhythms. In the case of periodic "time-of-day" aware automata we also observe a distinct *phase transition* during training, characterized by sudden and rapid discovery of the hidden time-dependent features, and we find that this transition coincides with a *bifurcation* in the RNN dynamics from which periodic orbits emerge. In this way, our work demonstrates how dynamical systems theory can provide insights into the *development* of representations learned by neural networks and illuminate pathologies encountered during training.

## 2 FRAMEWORK FOR LEARNING TIMED-AWARE BEHAVIOR WITH RNNS

In this section we introduce the automata-based learning framework that we use to study how RNNs learn time-aware behavior. Our framework focuses on *hidden* temporal variables that the RNNs must learn to model internally in their hidden layers. We discuss how these tasks are useful from the perspective of *developmental interpretability* of recurrent networks.

### 2.1 TEMPORAL AUTOMATA

**Temporal automata** (TA) are the basis for characterizing the time-dependence in our learning framework. Such an automaton is define by a tuple $(\Sigma, S, s_0, \Delta)$ where $\Sigma$ is a finite set of input symbols, $S$ is a finite set of states, and $\Delta : \Sigma \times S \times \mathbb{N} \to S$ is a *time-dependent* transition rule. The automaton starts in state $y_0 = s_0$ and undergoes a sequence of input-driven state transitions according to the recurrence relation $y_{t+1} = \Delta(y_t, u_{t+1}, t)$ where $u_t$ denotes the $t$-th input received by the automata. This definition extends the standard deterministic finite automata (DFA) by allowing the transition rule to change with time, thus enabling time-dependent state transitions.

We follow the clock-based formalism of *timed automata* (Alur & Dill, 1994) and define our transitions rules depend on time through binary functions of the clock values. Specifically, we introduce an underlying temporal variable $\Theta_t$ computed from the clocks of the timed automaton, and the overall the transition rule $\Delta$ can be described by two time-independent transitions rules $\delta_0, \delta_1 : \Sigma \times S \to S$ that are applied when $\Theta = 0$ and $\Theta = 1$, respectively: $\Delta(c, s, t) = \delta_{\Theta_t}(c, s)$.

**Periodic Timing** We first construct TA that emulates "time of-day"-aware behavior. For these machines, time is divided into contiguous days of $P$ timesteps, and each day is further divided into two phases of duration $P/2$, called the day(light) and night phases. The temporal variable then is defined to be a square wave — $\Theta(t) = 0$ if $t \pmod{P} < P/2$ and $\Theta(t) = 1$ otherwise — so the automaton uses the transition rules $\delta_0$ and $\delta_1$ during the day and night phases, respectively.

**Relative Timing** We also study a TA with transitions that depend on the amount of time between certain 'events'. Here, we introduce a "null" symbol $\phi$ that causes no change in state ($\Delta(\phi, s, t) = s$) and we equip these automata with a clock that tracks the number of timesteps since the last *non-null* symbol was received. The temporal variable $\Theta(t)$ indicates whether this clock exceeds a fixed threshold $\tau$: $\Theta(t) = 0$ if $\phi$ last appeared *at most* $\tau$ timesteps ago and $\Theta(t) = 1$ otherwise. The TA has higher probability of receiving the null symbol to ensure $\text{Prob}(\Theta = 0) \approx \text{Prob}(\Theta = 1)$.

## 2.2 TIMED AUTOMATA EMULATION TASKS

**TA Emulation by an RNN** For a TA $(\Sigma, S, \Delta, s_0)$, the sequence of symbols $u_1, \ldots, u_T$ will produce a sequence of states $y_1, \ldots, y_T$ according to the transition rule $y_t = \Delta(u_t, y_{t-1}, t)$. An RNN will respond to the same input sequence with a hidden state sequence $h_1, \ldots, h_T$ generated according to the update rule $h_t = F_h(u_t, h_{t-1})$, and model produces the associated output sequence $\hat{y}_t = F_y(h(t))$ where $F_h$ and $F_y$ are parameterizable functions that depend on the RNN architecture. The goal of training the network is to tune these functions such that $\hat{y}_t$ matches $y_t$.

**Training** of the RNNs is performed using a supervised learning approach. We first generate a dataset of input-output examples $\mathcal{D} = \{(u^i, y^i)\}$ of the TA. The input sequences are generated randomly with each symbol $u^i(t)$ drawn uniformly at random from the alphabet $\Sigma$ unless otherwise stated and the output sequences $y^i$ are computed from $u^i$ and the transition rule $\Delta$. We then train the RNN using stochastic gradient descent to learn the input-output mapping of sequences.

A **key concept** here is that the time-dependence of the TA transition rule $\Delta$ is hidden from the RNN. Whereas temporal variable $\Theta$ is an explicit input to the TA, the RNN only updates based on input symbols and its past hidden state, as shown in Figure 1b. The time-dependence of the transition rule cannot be inferred directly from the input sequence nor the output sequence individually; instead, it only when these two sequences are considered together that the time-dependence becomes clear. In this way, TA tasks are characterized by *temporal latent variables* that the model must discover and learn to represent through its hidden state sequence.

In defining tasks through automata, we know the temporal variables that the RNN must learn to emulate the TA behavior. In this way, we propose these TA tasks as a means of to interrogating both the *mechanisms* by which RNN represent temporal variables and the way in which these variables *develop* during training. Automata describe a wide range of capabilities, so this style of task offers an empirical playground for "opening the black box" of RNNs in many computational contexts.

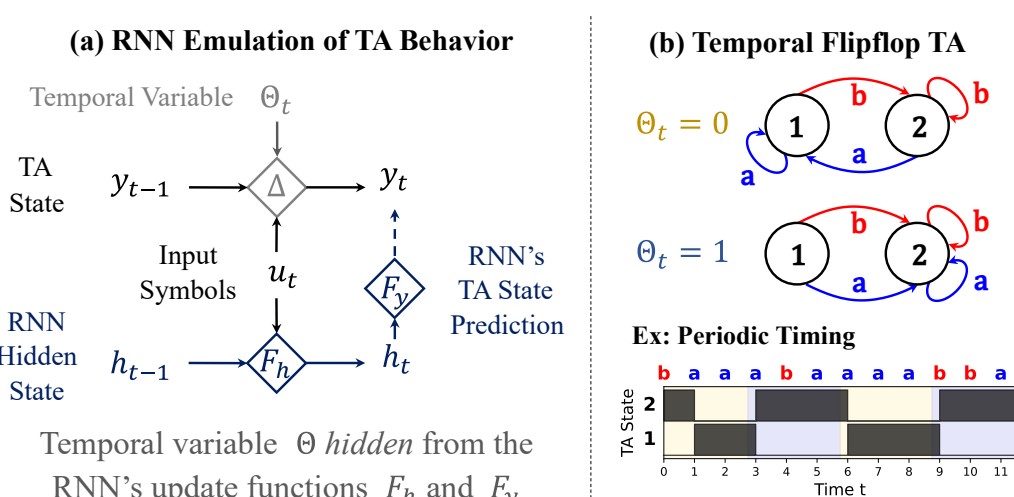

Figure 1: **(a) Hidden time-dependence of the TA Emulation Tasks**. The RNN receives a sequence of symbols $u_t$ as input which drives hidden state updates $h_t = F_h(h_{t-1}, u_t)$. From these hidden states the RNN must compute the correct TA state $y_t$ as its output $F_y(h_t)$. The *key concept* here is that the time-dependence of the TA transition rule $\Delta$ is hidden from the RNN. Whereas time is an explicit input to the TA, the RNN only updates based on input symbols and its past hidden state. The network must learn to represent the temporal information of the TA in the hidden state sequence. **(b) Temporal Flipflop TA** with states in $S = \{1, 2\}$. A directed edge *(i,j)* with label $l \in \Sigma = \{a, b\}$ denotes a transition to state $j$ when symbol $l$ is received in state $i$. Transitions into State 2 are the same for both $\Theta_t = 0$ and $\Theta_t = 1$, so they are time-*independent*. Transitions into State 1 are are time-*dependent* transitions, on the other hand, because they depend on $\Theta_t$.

### 2.3 FLIPFLOP MACHINES (STATE-*Independent* TA)

The TA studied in this paper resemble the flipflop automata from past mechanistic interpretability literature on RNNs (Sussillo & Abbott, 2009; Sussillo & Barak, 2013). Our (2-State) *Temporal Flipflop* TA in Fig. 1b has states $S = \{1, 2\}$ and input symbols $\Sigma = \{a, b\}$. The transition rules $\delta_0$ and $\delta_1$ for this TA are *state-independent*: Symbol $a$ causes the automaton to transition to State 1 when $\Theta = 0$, and it causes transitions to State 2 when $\Theta = 1$. Symbol $b$ causes the automaton to transition to State 2, regardless of the value of $\Theta$.

The TF exhibits both time-*independent* and time-*dependent* behavior — Symbol $b$ always induces transitions to State 2, but Symbol $a$ may lead to either States 1 or 2 depending on the value of the temporal variable $\Theta$. We therefore define two **evaluation metrics** for RNNs trained to emulate the flipflop TA — Time-dependent (TD) and Time-independent (TI) Accuracy, the accuracy of the network's predictions on timesteps when Symbol $a$ and Symbol $b$, respectively, was received as input. We continually compute the TD and TI accuracy of the network at every training iteration, and the resulting TI and TD learning curves are informative in monitoring the network's development.

## 3 LEARNING PERIODIC TIME-DEPENDENCE

We now present results collected when training RNNs to emulate the time-of-day aware temporal flipflop with $P = 24$ timesteps per day. We focus on single-layer Vanilla RNNs (Elman, 1990) with hidden layer of size $N_h = 64$. Refer to Appendix A for dataset and training details.

### 3.1 PHASE TRANSITION IN ACCURACY DURING LEARNING

For all networks, the time-independent and time-dependent learning curves followed the same phased structure shown in Fig. 2. Training begins with the network perfecting its time-independent behavior, while only slowly improving its time-dependent accuracy above 50% (equivalent to guessing). This plateau lasts for a few hundred gradient updates on average, after which the network undergoes a phase transition characterized by rapid improvement in its TD accuracy, rising from <55% to >90%. After the transition the network's time-dependent accuracy increases more slowly. Experiments using other recurrent architectures also produced phase transitions (see Appendix B), though the steepness of the transition was not as drastic.

Note that this phase transition is distinct from those caused by Grokking (Power et al., 2022). The sudden increasing in model performance occurs *simultaneously* for the train and test sets, so our models do not suffer from delay generalization. See Appendix C for data supporting this claim.

### 3.2 PERIODIC ORBITS ENCODING TIME OF DAY

We next sought to understand what causes the onset of learning the time-dependent behavior and how this is connected to the time-awareness required by the task. To this aim, we used techniques

Figure 2: **Phase Transition in accuracy** for RNNs trained to emulate the flipflop TA with period $P = 24$. The plots show the time-dependent (blue) and time-independent (red) learning curves for 30 random seeds, with one randomly selected seed drawn in bold. All of these networks show a clear *phase transition* in TD accuracy, which plateaus around slightly above 50% for hundreds of gradient updates before rapidly rising to >90%.

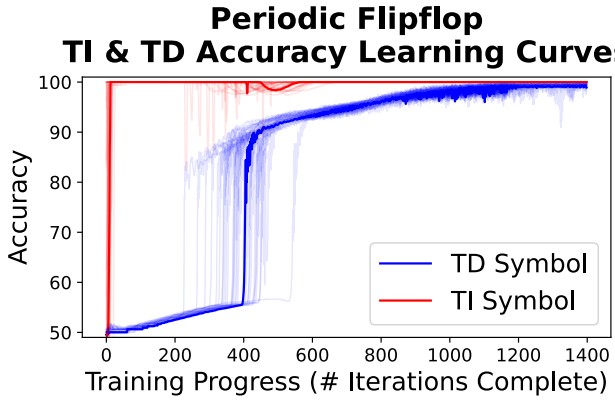

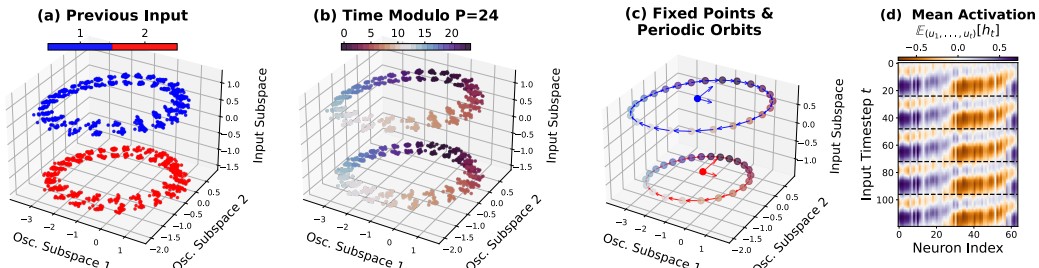

Figure 3: **Three dimensional dynamics** of an RNN trained to emulate the periodic TF. **(a)** & **(b)** show projections of the hidden state dynamics *during inference* on a few example input sequences. They have the same set of points, colored based on different features as indicated. One dimension encodes the previous input symbol to the RNN, whereas the dynamics in the other two dimensions form rings that encode time modulo $P = 24$. **(c)** These rings cluster around the periodic orbits induced by constant input strings ($a...a$ or $b...b$). **(d)** When viewed at the neuron-level, these three dimensional dynamics resemble waves traveling around the network in a circle.

from dynamical systems theory to explain (1) how the trained network's dynamics encode time of day and (2) how this representation emerges during the training processes.

We found the trained network uses three dimensions in its hidden state space to encode the two variables of the TA's transition rule — the previous input and the time of day. The former variable is encoded in one dimension, with positive and negative values indicating Symbols $a$ and $b$, respectively (Fig. 3a). The projection of the network's hidden states into the remaining two dimensions organize into point-clouds resembling rings, with position along the rings encoding time modulo $P = 24$ (Fig. 3b). In a sense, the RNN's hidden layer functions as a DFA whose states are pairs (previous input, time modulo $P$), where the DFA states are general regions of the hidden state space.

The rings themselves cluster about periodic orbits around *fixed points* (FPs) of the hidden layer. For a discrete driven dynamical system $h_t = F(h_t, u_t)$, a fixed point of the input $u$ is a state $h_*$ such that $h_* = F(h_*, u)$. Using the FP detection algorithm introduced by Sussillo & Barak (2013), we found that the networks all had a single *unstable* fixed point for each input symbol. Fig. 3c shows the (nearly) $P$-periodic hidden state trajectories induced by constant input (e.g. $a...a$ or $b...b$). Fig. 3d shows the mean activation $\mathbb{E}[h_t]$ by neuron by timestep, where the mean is taken over all sequences of input symbols $(u_1, u_2, ..., u_t) \in \Sigma^t$. One can we see that at the neuron level, the RNNs dynamics resemble periodic *waves* traveling through the network.

We found that this three dimensional subspace can be *directly* computed from the RNN weights. The input dimension is derived from the input weights $W_{uh} \in \mathbb{R}^{N_h \times 2}$ that project the input symbols, encoded via 1-hot vectors, into the hidden state. Based on the analysis by Di Marco et al. (2002), we found The two-dimensional time-of-day subspace is well approximated by the eigenvector of the top largest eigenvalue $|\lambda_{\max}|$ of $W_{hh}$. This eigenvalue was complex, and the real and imaginary parts of the associated eigenvector span a 2D space. Refer to Appendix D for further details.

Restricting the RNN's readout to seeing only these three dimensions has minimal effect on the model's post-training accuracy. If we make the output $\hat{y}_t$ depend on the *projection* of the hidden state $h_t$ into this subspace, the average TI accuracy remains 100%, and the average TD accuracy changes from 99.43% ± 0.29% (without the projection) to 99.45% ± 0.37% (with the projection). This result suggests these dimensions contain nearly all of the variables needed by the readout layers, though the mechanism by which the RNN maintains these variables is not presented here.

### 3.3 BIFURCATION DURING TRAINING

To understand how the input-dependent fixed points emerge through learning, we computed the input-dependent FPs at each training step, and we tracked the stability of these points through training by computing $|\lambda_{\max}|$, the absolute value of the largest eigenvalue of the Jacobian matrices $J_{kl} = \partial F_k(h, u)/\partial h_l$ at the FPs. Each FP is (locally) stable given constant input if $|\lambda_{\max}| < 1$.

**(a) Changing dynamics about Symbol A's FP during Training**

**(b) FP Stability through Training**   **(c) TI & TD Accuracy**

Figure 4: **(a) Bifurcation during training** at the fixed point for Symbol. A similar bifurcation is observed for the fixed point of Symbol B. **(b) & (c) Fixed point stability vs. TD Accuracy** throughout training. Refer to the main text immediately below for further details.

We found that, at the start of training, all networks had a single *stable* FP for each input symbol. Early in training, the networks respond to Symbol $b$ (TI) by moving its hidden state closer to the associated FP, near which the networks predicts TA State 2 with a probability close to 1. Symbol $b$ (TD) also takes the network close to a difference stable FP during the early stages. There, the networks' predictions are close to 0.5 probability for both TA states, i.e. it see it sees Symbol $b$ as causing random transitions. The network has not yet uncovered the hidden periodic variable, so it simply learns to predict the transition probabilities.

The two fixed points remain stable for much of the training process, and this stability results in a plateau in the TD accuracy. As training progresses, we see the emergence of decaying oscillatory dynamics in the hidden state dynamics given fixed input (see Fig 4). The largest magnitude eigenvalue $\lambda_{max}$ of the Jacobians at the FPs are complex at this point. The subfigures show the projections of the hidden state onto the real and imaginary parts of the associated eigenvectors.

The decay speed decreases as training progresses, which coincides with increasing $|\lambda_{max}|$ at the fixed point. $|\lambda_{max}|$ for each fixed point eventually crosses 1, making the associated fixed point *unstable*, and at this point we see the emergence of sustained oscillations (given constant input) about the fixed points. Note that theory guarantees that the fixed points are unstable for $|\lambda_{max}| > 1$, but this instability does not imply the existence of periodic orbits. Our empirical investigation of the networks' dynamics (Fig. 4) verifies these stable orbits exist.

The destabilization of the unstable FP appears to be correlated with the phase transition in the TD accuracy. In Fig. 4b-c, one can see the plateau in the TD accuracy ends precisely at the bifurcation from which stable periodic behavior emerges about the FPs. This result provides *quantitative* evidence that the cycles are the mechanism learned by the RNNs to encode time. A similar bifurcation is observed by Ribeiro et al. (2020) in their work on the vanishing and exploding gradient problem, though they do not report a connection between the loss and the bifurcation.

### 3.4 CHALLENGES OF LEARNING OSCILLATIONS VIA GRADIENT FLOW

We next tested whether beginning training with an RNN with stable periodic orbits would avoid the learning curve plateaus. In particular, we took RNN models trained to emulate the periodic flipflop

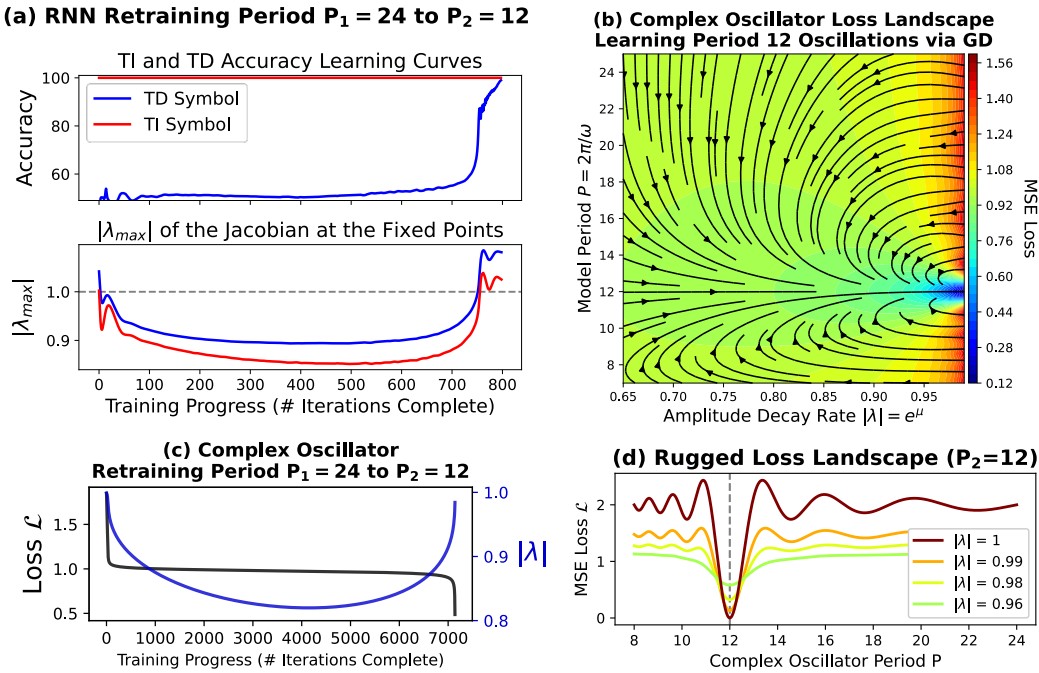

Figure 5: **Learning stable oscillations via gradient descent (a)** Learning curve of an RNN pre-trained to emulate the periodic flipflop TA with period $P_1 = 24$ and then re-trained to emulate the same automaton but with period $P_2 = 12$. The model starts with stable oscillations due to the pre-training, but upon re-training this stability is lost initially. **(b)** Phase portrait in the $(\mu, \omega)$ parameter space and **(c)** example trajectory of $|\lambda| := \exp(\mu)$ for a simple oscillator $\hat{y}_t = \exp[(\mu + i\omega)t]$ tuned to the target $y_t = \exp[i\omega_0 t]$ via gradient descent. Like the RNN, when the oscillator starts with stable oscillations, this stability is initially lost as the model adjusts its oscillation period. **(d)** The loss landscape has many local minima for $|\lambda|$ close to 1. Boundaries between these minima shrink as $|\lambda|$ decreases, so the oscillator must first reduce $|\lambda|$ to find a path to the global minimum.

with period $P_1 = 24$ and *re-trained* them to emulate the same automaton but with a different period $P_2 = 12$. The resulting learning curves of this re-training for and the associated fixed point stability curves are shown in Fig. 5a. Even though the model starts with stable periodic orbits ($|\lambda_{\max}| < 1$) of period $P_1 = 24$, these orbits *lose their stability* when the model commences training on period $P_2 = 12$. In losing stability, the model experiences another plateau in its TD learning curve, and it undergoes another phase transition when the periodic orbits become stable again.

To understand this loss of stability, we analyzed the simple task of approximating the continuous-time oscillation $y_t = \exp[i\omega_0 t]$ using the model $\hat{y}_t = \exp[(\mu + i\omega)t]$, where the parameters $\mu$ and $\omega$ are optimized via gradient flow to minimize the loss $\mathcal{L}_t = |y_t - \hat{y}_t|^2$ averaged over a time interval $[0, T]$. The frequency $\omega$ of the model's oscillations relates to the oscillation period as $\omega P = 2\pi$ and must be tuned to match the target frequency $\omega_0$. The parameter $\mu$ defines the decay/growth rate of oscillation amplitude, with $e^\mu$ being analogous to $|\lambda_{\max}|$ of the RNN fixed points. This parameter must converge to $\mu = 0$ (e.g. $e^\mu = 1$) to achieve stable oscillations.

The nonlinear dynamics of the gradient flow $(\dot{\mu}, \dot{\omega}) = (-\partial_\mu \mathcal{L}, -\partial_\omega \mathcal{L})$ can be understood from its phase portrait as in Fig. 5b. Here, the flow is slow when the oscillations are decaying ($\mu < 0$) because most terms in the loss $\mathcal{L}$ decay exponentially as $e^{2\mu T}$. This vanishing gradient is further exacerbated by the direction of the flow: like the RNN, trajectories of the complex oscillator approach the minimizer by first increasing the oscillation decay rate (decreasing $\mu$).

These roundabout trajectories through the $(\mu, \omega)$-space can be explained by the loss landscape given constant $\mu$ in Fig. 5d. For $\mu$ close to 0, the sinusoidal terms in the loss give rise to multiple local minimal with steeper boundaries between them as $\Delta\omega \to 0$. The model cannot escape these minima without first decreasing $\mu$ sufficiently to eliminate the walls between the minima. This simple model

suggests that vanishing gradients are not the sole reason for the plateaus observed in the RNN's learning curves; indeed, there is an addition ruggedness induced by the periodic nature of the task.

# 4 LEARNING RELATIVE-TIMING TEMPORAL FLIPFLOP

We now extend our analysis to the Relative-Timing TA with threshold $\tau = 5$ and a probability of seeing a non-null symbol $p = 0.2$. Here, we again observe three learning phases for the Relative-Timing TA as seen in Fig. 6. The time-independent behavior of Symbol $b$ is learned almost instantly in comparison to the learning of time-dependent behavior of Symbol $a$.

## 4.1 FIXED POINTS ENCODING TA STATES

Fig. 6 demonstrates the main difference between the relative timing and periodic Flipflop TA — the presence of *stable* fixed points post training. We find that there are in total four stable fixed points for the Relative-Timing TA—one for each symbol $\{a, b\}$ and two for the null-symbol $\phi$. Learning begins with just one stable fixed point for $\phi$, but as it progresses a second one appears as marked by the orange vertical dashed line. The emergence of this second stable fixed point indicates that the network has started learning its internal representation of time for this task since it coincides with the escape from the learning plateau and the beginning of the third phase of learning. The third phase is incremental in contrast to the rapid learning observed for the periodic TF because the network steadily learns to count up to the exact value of the threshold. This learning process is ubiquitous across all networks trained during our experiment with the variations being in the number of iterations for the second stable null-symbol fixed point to emerge.

## 4.2 EMERGENCE OF THE FIXED POINTS

The emergence of stable fixed points would indicate the lack of stable oscillations learned for the periodic timing

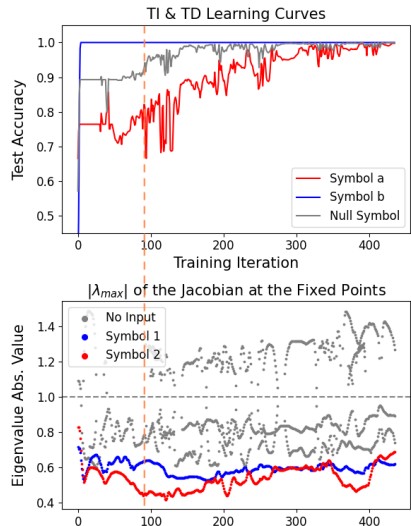

Figure 6: **The Relative-Timing 3-Phased Learning Process**. Learning dynamics are marked by one stable fixed point ($|\lambda_{max}| < 1$) for each symbol, and one stable null-symbol fixed point for each state—giving rise to $|\Sigma| + |S|$ stable fixed points. The learning starts with just one stable fixed point for the null-symbol, and the second one emerges around iteration 90 as marked by the orange vertical dashed line.

flipflop TA. We hypothesize that we obtain one stable fixed-point with the null-symbol for each state of the TA, and the non-null symbols cause the system to move *away* from these fixed points; the system uses the distance from these stable fixed points to encode the amount of time since a non-null symbol was received, and with each null-input received the system would get closer to these fixed points.

We visualize the RNN hidden state(s) in a lower dimensional subspace to validate this hypothesis. We use a 2-dimensional analysis in which the y-axis indicates some notion of phase/state of the TA and the x-axis encodes a representation of distance (or time) between points projected onto it. We select the y-axis to be the first principal component of the input weight matrix $W_{ih}$ of the last cell in the RNN network, and the x-axis to be the coefficients of a logistic regression model trained to classify whether a given hidden state is above or below threshold (two classes obtained by using the Relative-Timing TA train dataset as an oracle).

Fig. 7 plots this low-dimensional analysis of the RNN hidden state. The two stable fixed points associated with the null-symbol are marked with a cross sign on the right of the plot. The RNN is provided with the input sequence consisting of alternating Symbols $a$ and $b$ with $2\tau$ null-symbols between them, e.g. $a, \phi, ..., \phi, b, \phi, ..., \phi, a$ This allows us to study the convergence to stable fixed points and how the TA state behaves with each symbol as an input. We only change the color of the RNN state when a non-null symbol is received.

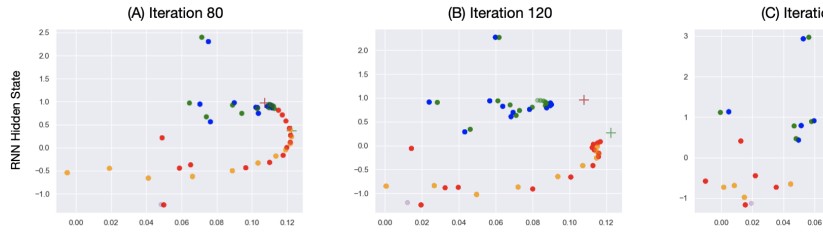

Figure 7: **Stable fixed points during training.** In **(a)** There is only one stable FP for the null-symbol and all RNN hidden states collapse to it. **(b)** shows the dynamics after the emergence of the second null-symbol stable FP. The dynamics/transitions are now learned and the network incrementally learns to count the latent variable threshold. **(c)** the latent variable threshold is learned, and it takes $\tau$ steps to collapse to the respective stable FP.

We observe that threshold $\tau$ timesteps are required for the RNN hidden state to collapse to one of the fixed points. The point to which it collapses depends on the current state of the TA network— (red cross for State 1, green cross for State 2). When provided with a non-null symbol, the RNN state is pulled away from the null fixed point and depending on which symbol was received, the RNN state is pulled to a specific point: the upper region (the green and blue points) are when Symbol $a$ is received and the lower region (the red and orange points) are for Symbol $b$.

During learning, when there is only 1 null-symbol fixed point (Fig. 7a), then there is always a collapse to this FP irrespective of the current state because the network has not yet learned to count. Fig. 7b represents the emergence of the second fixed point (pass the vertical orange line in Fig. 6c) and hence there is an increase in the accuracy. However, although the fixed point structure that enables countable has been learned at this point, the network still needs to fine-tune its ability to count to match threshold $\tau$ of the TA, which the model completes by the end of training (Fig. 7c)

## 5 DISCUSSION

In this paper, we use tools from dynamical systems to explain how recurrent neural networks represent time in their hidden states, and we present evidence to illuminate how these internal representations develop during training. We accomplish this by defining a new family of automata-based time-aware sequence modeling tasks. For the state-independent automata we call Temporal Flipflops (TF), we find that the networks learn reusable behaviors of time that improve learning and generalization. Our experiments show that a common three-phase learning structure is observed independent of the form of time-dependence, but we observed differences in the emergent temporal representations and the bifurcations driving phase transitions during training.

Phase transitions during training have been observed in many ML applications (Power et al., 2022; Murakami & Shono, 2022; Tang et al., 2021; Vecoven et al., 2021), so this style of developmental analysis may provide insights into the training pathologies in other contexts. The plateauing phenomena is closely related to vanishing and exploding gradients, a fundamental challenge in learning long-term dependencies with RNNs. There is a long-standing hypothesis attributing gradient pathologies during training to bifurcations in RNN dynamics (Doya, 1993; Bengio et al., 1994; Pascanu et al., 2013), and to our knowledge, this paper is one of the first to demonstrate this connection empirically for trained RNNs with 1000s of parameters. Future studies using dynamical systems to analyze model development may prove informative to designing improved recurrent architectures.

There is no guarantee that fixed point analysis through the training process will be applicable to recurrent models trained on other tasks, even those described by timed automata. For this form of analysis to be applied more generally, there is a need for more efficient methods of tracking the fixed points through training, for instance by modifying the learning process (Smith et al., 2021). Still, even without these developments, we hope our work conveys the message that we can learn much about RNNs by studying the dynamics of learning process itself, in an aim for safer and more interpretable models.

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

## SUPPLEMENTAL MATERIAL

## A    EXPERIMENT DETAILS

### A.1    ARCHITECTURE

We used single layer Vanilla RNNs for both flipflop automata tasks. Given an input sequence $u = (u_1, u_2, ..., u_T)$ with each $u_t \in \mathbb{R}^{N_{in}}$, an Elman RNN produces the output sequence $y = (y_1, ..., y_T)$ with $y_t \in \mathbb{R}^{N_{out}}$ following the equations

$$h_{t+1} = \tanh(W_{hh}h_t + W_{uh}u_{t+1} + b_h) \quad , \quad y_t = \sigma(W_{hy}h(t) + b_y) \tag{1}$$

Here $W_{uh} \in \mathbb{R}^{N_h \times N_{in}}$, $W_{hh} \in \mathbb{R}^{N_h \times N_h}$, and $W_{hy} \in \mathbb{R}^{N_{out} \times N_h}$ are the input, hidden, and output weights, respectively. These parameters were initialized with each component randomly drawn from a normal distribution with mean 0 and standard deviation $1/\sqrt{N_{in}}$, $1/\sqrt{N_h}$, $1/\sqrt{N_{out}}$ respectively.

$N_h$ is the dimension of the hidden state space, which we set to $N_h = 64$ or $N_h = 32$ for the periodic and relative timing temporal flipflop experiments, respectively.

The parameters $b_h \in \mathbb{R}^{N_h}$ and $b_y \in \mathbb{R}^{N_{out}}$ are bias vectors. These vectors were also initialized with each component randomly drawn from a normal distribution with mean 0 and standard deviation $1/\sqrt{N_h}$ and $1/\sqrt{N_{out}}$ respectively.

The initial hidden state $h(0) \in \mathbb{R}^{N_h}$ for each model was also trained parameter. The components of these vectors were initially drawn from a normal distribution with mean 0 and standard deviation $g_{h_0}/\sqrt{N_h}$ with $g_{h_0} = 0.05$.

### A.2    TASK PARAMETERS

For both tasks, we generated a dataset with 32768 training examples and 4096 test examples. Each example was a pair of sequences $(u, y)$ with sequence length $T$ where $u = (u_1, u_2, ..., u_T)$ is a sequence of TA input symbols and $y = (y_1, y_2, ..., y_T)$ is the associated TA output sequence. The flipflop TA studied in this paper both had two input symbols, which where represented as vectors using 1-hot encoding: $u(t) = [1, 0]^\top$ to indicate Symbol A and $u(t) = [0, 1]^\top$ to indicate Symbol B. These TA also had two states, which we represented using a single binary value: $y_t = 0$ and $y_t = 1$ States 1 and 2 of the TA, respectively.

### A.3    TRAINING PARAMETERS

We used PyTorch's (Paszke et al., 2019) implementation of Vanilla RNNs (the RNNCELL class in particular). The models were trained using Adam optimization (Kingma & Ba, 2014) for the periodic flipflop TA and RMSProp (Graves, 2013) for the relative timing TA. Both tasks used binary cross entropy loss, but similar results were obtained for MSE loss as well.

We manually tuned training hyperparameters for each task. The values used for our experiments are listed in Table 1.

|  | Periodic Timing | Relative Timing |
|---|---|---|
| Input & output dimensions $N_{in}, N_{out}$ | 2,1 | 2,1 |
| Hidden dimension $N_h$ | 64 | 32 |
| Input probability $p$ | 1.0 | 0.2 |
| TA period $P$ | 24 | — |
| TA timing threshold $\tau$ | — | 10 |
| Training sequence length $T$ | 168 | 60 |
| Max training iterations | 1500 | 512 |
| Learning rate | $2 \cdot 10^{-4}$ | $5 \cdot 10^{-4}$ |
| (Mini)batch size | 128 | 128 |
| RNN Optimizer | Adam | RMSProp |

Table 1: Architecture, Task, & Training Parameters
NOTE: A training *iteration* is different from an *epoch*, which consists of as many training iterations required to loop through the dataset once. For the periodic flipflop task, with a batch size of 128 and a training dataset with $10,000$ examples, each epoch consists of 78 training iterations. Our models require approx. 200 epochs to converge.

## A.4 FIXED POINT COMPUTATION

We used our own implementation of fixed point finding algorithm for RNNs introduced by Sussillo & Barak (2013). An implementation of their algorithm exists for the Tensorflow machine learning library (Golub & Sussillo, 2018), but we found it easier to implement our own version of the code than to transition our work from PyTorch to Tensorflow.

## B PERIODIC FLIPFLOP: LEARNING CURVES OF OTHER ARCHITECTURES

We also trained two other recurrent architectures to emulate the periodic flipflop TA: (1) Cayley (Orthogonal) RNNs (Helfrich et al., 2017) and (2) Lipschitz RNNs (Erichson et al., 2020). These architectures are good candidates to learn stable periodic behavior due to the additional structure imposed on their hidden weights to alleviate issues with vanishing gradients.

The TA task for these models had period $P = 10$ timesteps, sequences of length $T = 60$, with 10k train sequences and 5k validation sequences. All models had a single layer of $N_h = 32$ hidden units, and they were trained using Adam (Kingma & Ba, 2015) with learning rate $10^{-3}$ and mini-batch size 32.

For the Lipschitz RNN we found that the hyperparameters $\epsilon = 0.9$, $\beta = 0.7$, and $\gamma = 0.001$ resulted in models that learned efficiently. We set their recurrent weights to zero initially because this choice resulted in the most efficiently trained models. Similar phase transitions were observed when these recurrent weights were initially drawn from a Gaussian distribution with mean zero and stdev $1/\sqrt{N_h}$, but we found this standard choice of initialization caused these models to converge less quickly.

Fig. S1 below plots the mean $\pm$ 1 stdev time-dependent accuracy learning curves for these models, averaged over 10 training seeds. The three models shown here — Vanilla RNN, Cayley RNN, and Lipschitz RNN — all have clear phases of their learning process. Like the Vanilla RNNs, the Cayley RNNs plateaus between 50% and 60% TD accuracy initially and sees a more rapid increase in this metric later during training. This phase transition is not as steep as the transition for the Vanilla RNN models, but there is still a clear inflection point in the learning curves.

The learning curves of the Lipschitz RNN (Fig. S2) do not necessarily plateau; rather, these models initially improve at a slow but consistent rate, and after approx. 200 gradient updates this rate of improvement increases suddenly. This behavior might be characterized by a second order or continuous phase transition.

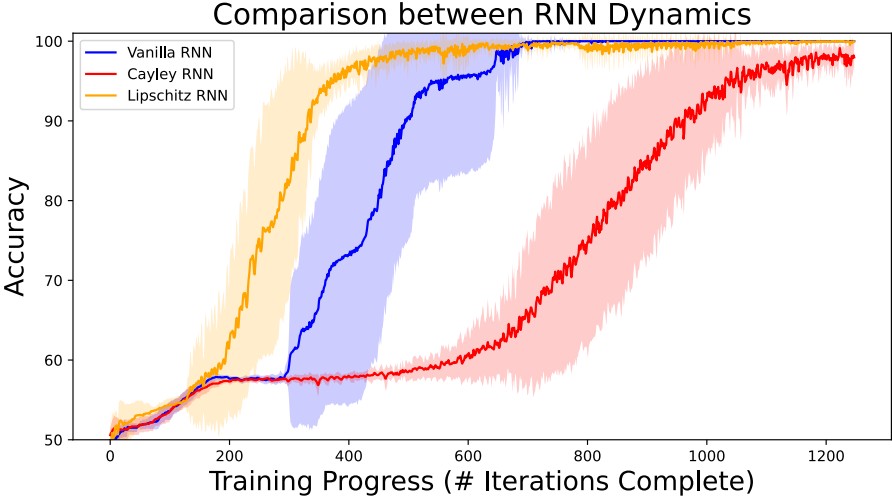

Figure S1: **TD accuracy learning curves for other recurrent architectures** for the periodic flipflop TA with $P = 10$. The plot shows the mean ($\pm$ 1 stdev) TD accuracy vs. training iteration, averaged over 10 seeds.

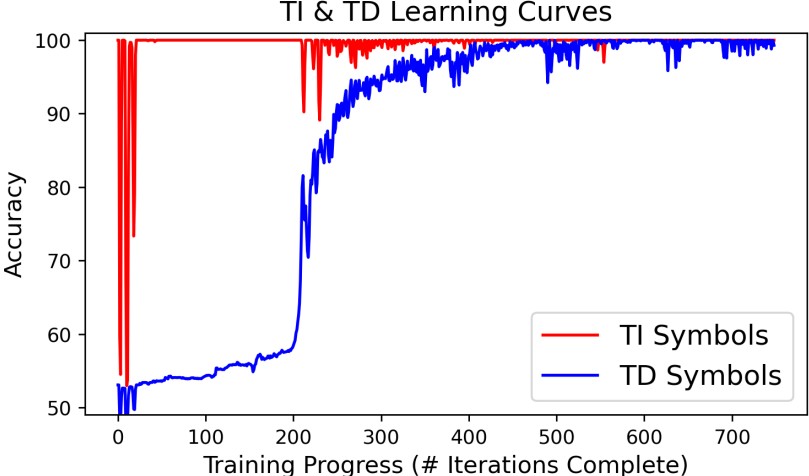

Figure S2: **Lipschtiz RNN Example TI and TD accuracy learning curves** for the periodic flipflop TA with $P = 10$.

## C   Periodic Flipflop: Train vs. Validation Accuracy

Phase transitions during training commonly coincides with *grokking* — delayed generalization well after a model has learned the training dataset (Power et al., 2022). To support our claim that grokking is *not* the present in our experiments, Fig. S3 below plots the difference in train and validation time-dependent (TD) accuracy for 10 Vanilla models trained to emulate the periodic flipflop TA with $P = 24$. These models were trained in the same way as those whose data was presented in the main paper, just with different random seeds. The difference in between these two metrics is small, never exceeding 0.5% at any point during training, so we see no indication of delayed generalization in these networks.

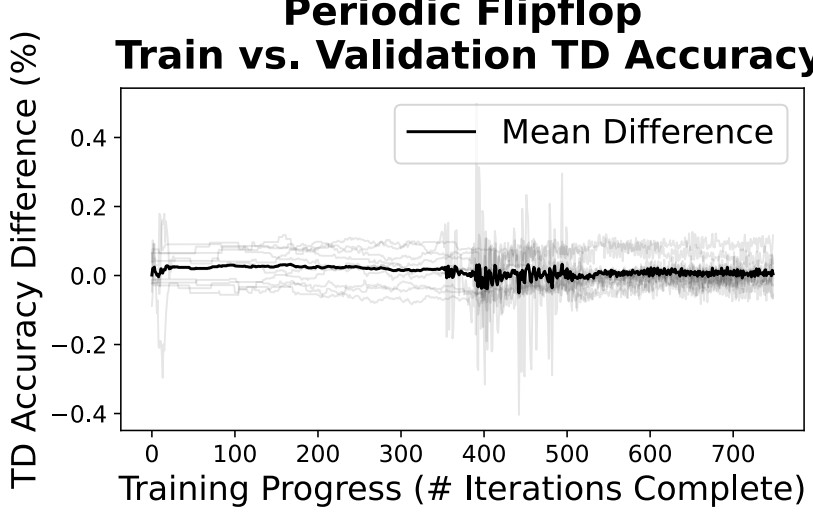

Figure S3: **Comparison between the train and validation TD accuracy** for the periodic flipflop TA. This figure shows the difference (train − validation) in time-dependent (TD) accuracy at each training iteration for 10 networks trained to emulate the $P = 24$ periodic flipflop TA. These networks were trained via the same script as those whose results we presented in the main paper, with the only difference being the seed for the experiments. The difference in between these two metrics is small, never exceeding 0.5% at any point during training, so we see no indication of grokking in these networks.

## D   Periodic Flipflop TA: Dimensionality Reduction

In the main paper, for RNNs trained on the periodic flipflop, we project their 64-dimensional hidden state dynamics into three dimensions to allow for visualization of their learned representation. Below are the details our of dimensionality reduction algorithm.

The dimension $b_u \in \mathbb{R}^{N_h}$ encoding the network's previous input follows directly from the input weights $W_{uh} \in \mathbb{R}^{N_h \times 2}$ that project the input symbols, encoded via 1-hot vectors, into the hidden state. We found that weights associated with each symbol (i.e. the rows of $W_{uh}$) are nearly anti-parallel, so we took $b_u$ to be the unit vector in the direction of $(\text{row}_1(W_{uh}) - \text{row}_2(W_{uh}))/2$.

The time-of-day dimension can be computed directly from the hidden-to-hidden weights $W_{hh}$. Based on the analysis by Di Marco et al. (2002), we found these dimensions are well approximated by the eigenvectors of the top largest eigenvalue $|\lambda_{\max}|$ of $W_{hh}$. This eigenvalue was complex, so the real and imaginary parts of its associated eigenvector span a two-dimensional subspace of the hidden state space.

Note that a similar result can be achieve simply by using principal component analysis (PCA) of the hidden state trajectories (c.f. results in Appendix G). The input and oscillatory subspaces are

not necessarily align aligned when performing PCA, however, and we felt that constructing these subspaces from the learned parameters of the RNN constituted a stronger result.

# E    LOSS FOR THE COMPLEX OSCILLATOR TASK

In Section S3.4, we analyzed the simple task of approximating the continuous-time oscillation $y_t = e^{i\omega_0 t}$ using the model $\hat{y}_t = e^{(\mu + i\omega t)}$, where the parameters $\mu$ and $\omega$ are optimized via gradient flow to minimize the loss $\mathcal{L}_t = |y_t - \hat{y}_t|$ averaged over a time interval $[0, T]$. Below is a calculation of the integral, to support the claim that most terms in the loss decay exponentially as in $\mu$.

$$\mathcal{L} = \frac{1}{T}\int_0^T |y_t - \hat{y}_t|^2 dt = \frac{1}{T}\int_0^T \left|e^{i\omega_0 t} - e^{(\mu+i\omega)t}\right| dt^2$$

$$= \frac{1}{T}\int_0^T \left[e^{i\omega_0 t} - e^{(\mu+i\omega)t}\right]\left[e^{-i\omega_0 t} - e^{(\mu-i\omega)t}\right] dt$$

$$= \frac{1}{T}\int_0^T \left[1 - e^{[\mu+i(\omega-\omega_0)]t} + e^{[\mu-i(\omega-\omega_0)]t} + e^{2\mu t}\right] dt$$

$$= 1 + \frac{1}{T}\left[\frac{e^{[\mu+i(\omega-\omega_0)]t}}{\mu - i(\omega-\omega_0)} + \frac{e^{[\mu+i(\omega-\omega_0)]t}}{\mu + i(\omega-\omega_0)} - \frac{e^{2\mu t}}{2\mu}\right]\Bigg|_0^T$$

$$= 1 - \frac{e^{2\mu T} - 1}{2\mu T} + \frac{1}{T}\left[\frac{e^{[\mu+i(\omega-\omega_0)]T} - 1}{\mu - i(\omega-\omega_0)} + \frac{e^{[\mu+i(\omega-\omega_0)]T} - 1}{\mu + i(\omega-\omega_0)}\right]$$

Simplifying the remaining term with complex values and making the substitution $\Delta\omega = \omega - \omega_0$,

$$\frac{e^{[\mu+i(\omega-\omega_0)]T} - 1}{\mu - i(\omega-\omega_0)} + \frac{e^{[\mu+i(\omega-\omega_0)]T} - 1}{\mu + i(\omega-\omega_0)}$$

$$= \frac{(\mu + i\Delta\omega)\cdot\left(e^{[\mu-i\Delta\omega]T} - 1\right) + (\mu - i(\omega-\omega_0))\cdot\left(e^{[\mu+i\Delta\omega]T} - 1\right)}{\mu^2 + \Delta\omega^2}$$

$$= \frac{2\,\mathrm{Re}\left[(\mu - i\Delta\omega)\cdot\left(e^{[\mu+i\Delta\omega]T} - 1\right)\right]}{\mu^2 + \Delta\omega^2}$$

$$= \frac{2\,\mathrm{Re}\left[(\mu - i\Delta\omega)\cdot\left(e^{\mu T}\cos(\Delta\omega T) - 1 + e^{\mu T}\sin(\Delta\omega T)\right)\right]}{\mu^2 + \Delta\omega^2}$$

$$= 2\,\frac{\mu\left(e^{\mu T}\cos(\Delta\omega T) - 1\right) + \Delta\omega e^{\mu T}\sin(\Delta\omega T)}{\mu^2 + \Delta\omega^2}$$

This gives us the overall loss

$$\mathcal{L} = 1 - \frac{e^{2\mu T} - 1}{2\mu T} + \frac{2}{T}\,\frac{-\mu + e^{\mu T}\left(\mu\cos(\Delta\omega T) + \Delta\omega\sin(\Delta\omega T)\right)}{\mu^2 + \Delta\omega^2}$$

# F    RELATIVE-TIMING FLIPFLOP: DIMENSIONALITY REDUCTION

In order to visualise the hidden states of the network trained on the relative-timing TA and validate our hypothesis, we project into a two-dimensional space. We aim to have the y-axis represent the state of the TA and the x-axis encode a representation of distance (or time) between points.

Accordingly, the y-axis is constructed by conducting PCA and selecting the first principal component of the weight matrix $W_{ih}$ multiplied to input vectors of the last RNN cell. We call this vector $h_{TA} \in R^d$ since it encodes the RNN representation of the TA state, where $d$ is the size of the RNN hidden state. To obtain the x-axis and represent distance, we train a logistic regressor to classify RNN hidden states into binary outputs corresponding to the latent variable $\Theta_{\mathrm{relative}}$. Training a classifier provides an implicit measure of distance from the boundary between classes, something we

notice when we project our points in lower dimensional spaces. We call the weights of the classifier $h_{lr} \in R^d$.

We now want to project the RNN hidden states into $Span\{h_{TA}, h_{LR}\}$. To do so we construct matrix $H = [h_{TA}^T, h_{LR}^T] \in R^{d \times d}$.

Using linear algebra, the projection of a given hidden state $h$ can be obtained by $(w_{TA}, w_{lr}) = H^+ h$ where $H^+ = (H^T H)^{-1} H^T$ and $(w_{TA}, w_{lr})$ are the desired coordinates in the span of the vectors.

## G  PERIODIC FLIPFLOP: DYNAMICS ANALYSIS FOR PERIOD $P = 10$

In this appendix we include additional results from our experiment involving the periodic flipflop automaton. These results were generated using the same procedure but for period $P = 10$ and sequence length $T = 60$.

### G.1  POST-TRAINING DYNAMICS

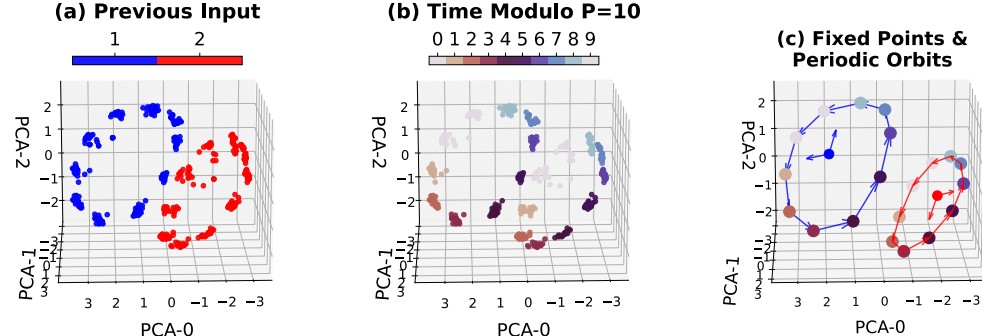

Figure S4: **PCA Visualization of the dynamics of an RNN trained to emulate the periodic TF —** a second example, akin to Fig. 3. This figure shows the post-training dynamics of a network trained using the same procedure and parameters described in Section B of the Supplemental Material, just with a different random seed.
(a) & (b) show the top three principal components of the hidden state dynamics *during inference* on a few example input sequences. They have the same set of points, colored based on different features as indicated. (c) shows the periodic orbits encoding time of day, with each state colored based on time modulo $P = 10$.
The PCA of the trained network's dynamics reveals its hidden states during inference organize into two point-clouds resembling rings. These rings encode the two relevant pieces of information required by the network to predict the current TA state: the previous input symbol and the time of day. The rings themselves encode the former data, as indicated in subplot (a), whereas the position along the rings encodes time modulo $P = 10$ (subplot b).
The rings themselves cluster about periodic orbits around fixed points (FPs) of the hidden layer. After training, we found that the networks all had a single unstable fixed point for each input symbol, indicated by the blue and red points in subplot (c), which also shows shows the hidden state trajectories induced by constant input strings (e.g. $a...a$ or $b...b$).

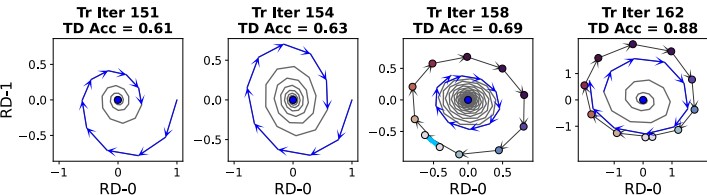

Figure S5: **Changing dynamics nearby the fixed point for Symbol A during training** — a second example, akin to Fig. 4.

We found that all trained networks initially have a single *stable* FP for each input symbol. As training progresses, we see the emergence of decaying oscillatory dynamics in the hidden state dynamics given fixed input, as shown in the left two plots of this figure. The highest magnitude eigenvalue $\lambda_{\max}$ of the Jacobians at the FPs are complex at this point. The plots show the projections of the hidden state onto the real and imaginary parts of the associated eigenvectors.

The decay speed decreases as training progresses, which coincides with increasing $|\lambda_{\max}|$ at the fixed point. This largest eigenvalue for each fixed point eventually crosses 1 in absolute value, making the associated fixed point unstable, and at this point we see the emergence of sustained oscillations (given constant input) about the fixed points. These trajectories are quasi-periodic, but approach period $P$ as training progresses.

## G.2   TIME-DEPENDENT ACCURACY VS. FIXED POINT STABILITY

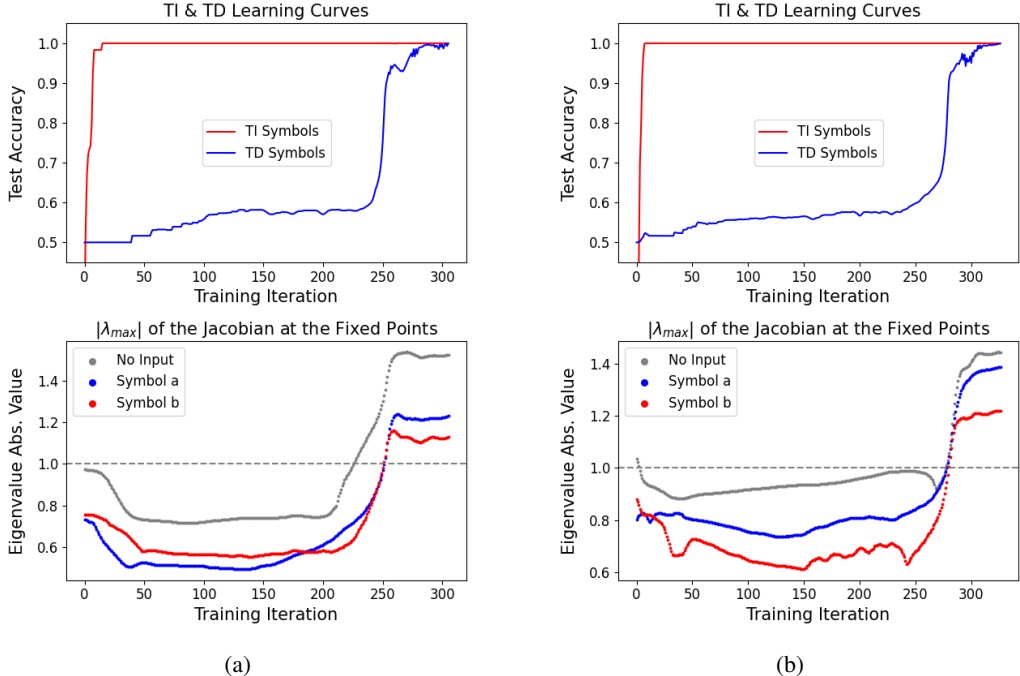

(a)                                        (b)

Figure S6: **Relationship between fixed point stability and TD Accuracy throughout training** — further examples for RNNs trained with sequence length $T = 60$. (a) and (b) refer to two different networks trained using the same procedure and parameters described in Section B of the Supplemental Material, just with different random seeds.

As in Section 3, the time-independent and time-dependent learning curves (top row of plots) for these networks followed the same three-phased structure. Training begins with the network perfecting its time-independent behavior, while only slowly improving its time-dependent accuracy above 50% (equivalent to guessing). This phase lasts for at least half of the training process (on average), after which the network undergoes rapid improvement in its TD accuracy, rising from <65% to >90% in a fraction of the duration of the first phase. The final phase of learning is characterized by a slower convergence of the network's time-dependent accuracy to >99%.

The bottom row of plots tracks of the modulus $|\lambda_{\max}|$ of the largest eigenvalue $\lambda_{\max}$ for all fixed points (FPs) of the networks. Each of these networks had a *single* fixed point associated with each type of input — Symbol A, Symbol B, and the "Null Symbol" (the zero vector). The FPs associated with Symbols A and B are initial *stable* fixed points because their $|\lambda_{\max}| < 1$, but they *destabilized* later in training when their $|\lambda_{\max}|$ crosses the threshold of 1. The destabilization happens at the same time for both Symbol A and Symbol B here, though this result does not hold in general.

The destabilization of the FPs of Symbols A and B appears to be correlated with the phase of rapid learning of the TD accuracy. One can see the plateau in the TD accuracy ends around when the bifurcation point for the TD symbol.

The Null Symbol also has a single fixed point for both of these networks. This fixed point is stable early in training and becomes unstable as training progresses, but the point of destabilization does not always coincide with spikes in the network's TD accuracy.

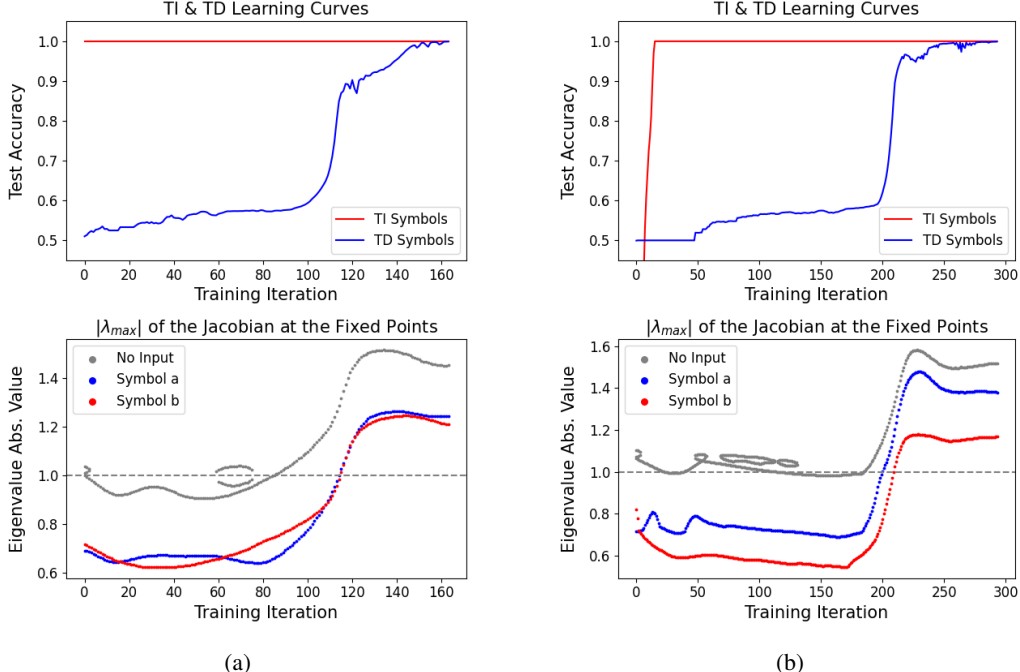

(a)                                                        (b)

Figure S7: **Relationship between fixed point stability and TD Accuracy throughout training** —
further examples for RNNs trained with sequence length $T = 60$. (a) and (b) refer to two different
networks trained using the same procedure and parameters described in Section B of the Supple-
mental Material, just with different random seeds.

The learning curves for these plots are qualitatively the same as those in the figure on the previous
page. Similarly, the correlation between the increase in the networks' time-dependent accuracy (top
row, in blue) and the moduli $|\lambda_{\max}|$ of the largest eigenvalue of the fixed point (FP) associated with
Symbol A (bottom row, in blue).

Here, we do see a difference in the structure of the fixed points associated with Null Symbol (bot-
tom row, in gray): For both of these networks, a pair of fixed points associated with the null symbol
appears during training and vanishes before the networks' converge. This result may not be illumi-
nating for the periodic flipflop task, in which the network was never presented with the null symbol
as input.

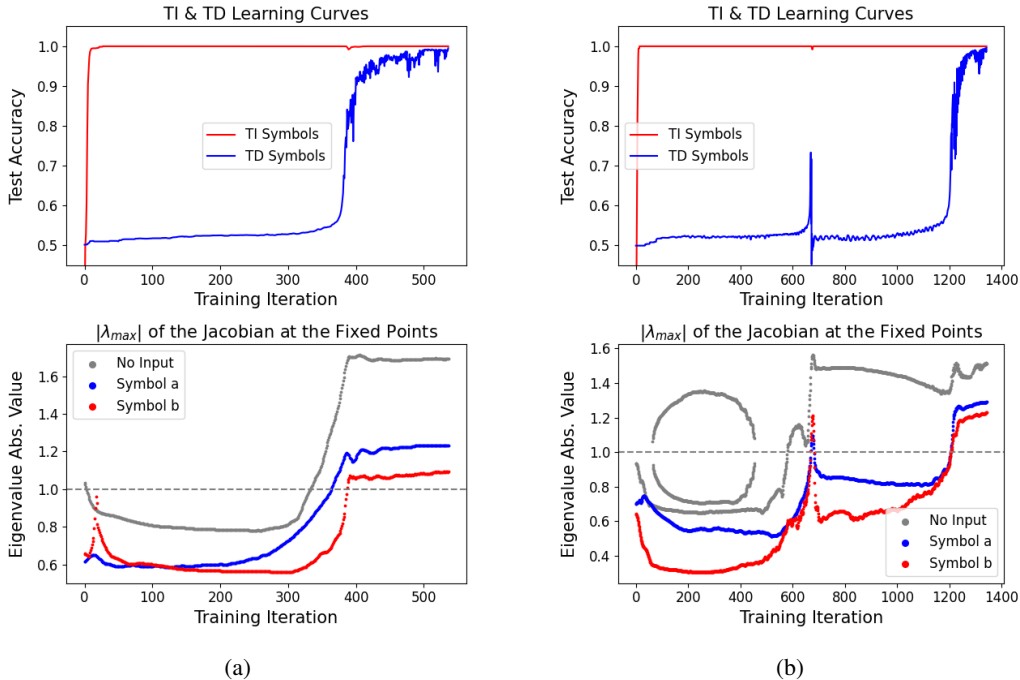

Figure S8: **Relationship between fixed point stability and TD Accuracy throughout training** — examples for RNNs trained with sequence length $\mathbf{T} = \mathbf{180}$. (a) and (b) refer to two different networks trained using the same procedure and parameters described in Section B of the Supplemental Material, just with $T = 180$, more training iterations, and different random seeds.

The learning curves for these plots are qualitatively the same as those in the figures on the previous two page. Quantitatively, these networks trained on longer sequence lengths require more training iterations to converge, which is reasonable because these models were evaluated over longer periods of time. In **(a)**, we see a similar correlation between between the increase in the network's time-dependent accuracy (top row, left, in blue) and the modulus $|\lambda_{\max}|$ of the largest eigenvalue of the fixed point (FP) associated with Symbol A (bottom row, left, in blue).

The learning process is more complex in **(b)**. The network's time-dependent accuracy plateaus around 0.5 initially and starts to rise from the plateau around training iteration 640. This escape from the plateau is short-lived, however, as the network quickly returns back to the plateau for training iterations 750 through 1200 before escaping a second time and converging to perfect accuracy. The plot of $|\lambda_{\max}|$ for Symbols A and B for the network in (b) also show a similar trend: the FPs are initially stable, become unstable temporarily when the network's TD accuracy spikes for the first time, but become stable again after the TD accuracy drops back to the plateau. This result further supports this correlation between the TD accuracy and the stability of the FPs of the network.

In (b), we also see another example of the null symbol having multiple fixed points that vanish before the end of the training.

