# OpenReview forum: "On the Dynamics of Learning Time-Aware Behavior with RNNs"
_ICLR.cc/2024/Conference — Submitted to ICLR 2024_

### Official Review · Reviewer_oza4 · 2023-10-29

**Soundness:** 3 good
**Presentation:** 3 good
**Contribution:** 2 fair
**Rating:** 3
**Confidence:** 3

**Summary:**

The authors attempt to explain how recurrent neural networks represent time in their hidden states and how the internal representations develop during training from the perspective of dynamical system. Specifically, they use Temporal Flipflops (TF), automata-based
time-aware sequence, to find we find the networks to learn reusable behaviors of time that improve learning and
generalization. They provide numerical studies to show that the learning process has a common three-phase pattern.

**Strengths:**

•	The author delves into the interpretability of RNNs, aiming to comprehend and monitor the hidden representation of RNNs. This offers the research community a more transparent insight into RNN model operations.

•	The author includes meticulous derivations/explanations of dimension reduction using PCA and the loss for the complex oscillator task in the appendix section, reinforcing the paper's mathematical foundation.

•	The observation of the common three-phase learning structure as time-independent offers a novel perspective on capturing the temporal dependencies of time series.

**Weaknesses:**

•	The scenario tested is very simple and hence generalization of the findings is questionable.

•	While the author elucidates the statistical aspects comprehensively, a more detailed explanation of the autoencoder's use would have been beneficial. For instance, discussing the motivations behind its selection, why it's deemed the best choice, or testing its performance against alternatives like variational autoencoders.

•	In the experiment section, the author evaluates the results using five datasets from the Monash Time Series Forecasting Repository based on dimensions, frequencies, and length. The chosen samples appear to be of a small size. Merely calculating the average rank might not provide an unbiased and comprehensive evaluation. The results would be more persuasive if the author utilized a broader range of datasets and presented a critical difference plot.

•	The author might consider employing other techniques, such as artificially creating uneven sampling frequencies, to garner more samples.

**Questions:**

•	Why using autoencoders rather than other deep learning models such as CNN to estimate the probability distributions?

•	How to capture the dimensional dependency using this architecture?

•	Why do we take this particular subset of datasets?

•	Are the blue and the red dots with two arrows the fixed points for the two states?

•	In Fig 7 (a), the caption says there is a single stable fixed point, while there are two +'s. Which one of the two is the fixed point you meant?

---

### Official Review · Reviewer_HfNN · 2023-10-30

**Soundness:** 3 good
**Presentation:** 3 good
**Contribution:** 3 good
**Rating:** 3
**Confidence:** 2

**Summary:**

The paper proposed the learning of timed automata (TA) and "neural flip-flops" to learn time-dependent dynamics using RNNs.
The paper shows that RNN can learn stable cyclic dynamics when trained on time-dependent data, and that these dynamics can become unstable if the model is finetuned on data with different period times.

**Strengths:**

- The paper is easy to understand
- The paper studies several phases of training (beginning, later stages, finetuning), which is quite rare and a great contribution here.

**Weaknesses:**

- My main concern about the work is that the tasks considered in this paper are rather artificial or synthetic. It is unclear how the results transfer to real-world datasets, which are more nuanced and may contain a superposition of multiple overlapping periods.
- Another weak point of the work is how the observed results translate into building better RNN models and learning algorithms. Specifically, the paper does a great job characterizing the dynamics but how can this information be used to improve our methods
- Finally, I think the paper would be stronger if it also considered more modern RNN architectures. Particularly, there have been some breakthroughs in the past few years (see Rusch et al. 2021, 2022, and Gu et al. 2021).

**References**.
Gu et al. 2021, Efficiently Modeling Long Sequences with Structured State Spaces.
Rusch et al. 2021, Unicornn: A recurrent model for learning very long time dependencies.
Rusch et al. 2022, Long expressive memory for sequence modeling.

**Questions:**

What are the real-world implications of the observed results for building better ML models and algorithms?

---

### Official Review · Reviewer_kx7S · 2023-10-31

**Soundness:** 3 good
**Presentation:** 1 poor
**Contribution:** 2 fair
**Rating:** 3
**Confidence:** 3

**Summary:**

This paper performs an empirical evaluation for understanding how RNNs represent time in their hidden states. Specifically, the paper analyses the dynamics of RNNs both during and after training with the help of a set of supervised learning tasks designed with timed automata (TA). The experiments show several interesting phenomena such as a common three-phased learning structure independent of dependence on time as a variable.

**Strengths:**

- The paper presents an interesting empirical study seeking to understand how time is represented internally in the RNN hidden states, using tools from formal automata design.

- The experiments shed light on how the hidden states evolve both during training, as well as what they represent after the training is completed.

- The experiments are cleanly divided into different categories, each testing specific hypotheses. The tasks for time automata emulation all share the same structure (seq-to-seq mapping supervised learning tasks) which makes it easy to evaluate the different results in a unified setting.

**Weaknesses:**

- The paper is extremely difficult to read and understand because most of the sections do not include a broad high level context and directly dive into technical details with math/notations. In order to make the results widely accessible to the machine learning community of ICLR, most sections of the paper need to be re-written with appropriate context.

- The related works is minimal /missing. Several prior works that have attempted to understand neural network dynamics, and in particular those that have tried to understand the dynamics of recurrent models need to cited, and discussed. It will help to have a separate related works sections for this.

- The experiments are on very toy settings. While they are helpful in understanding each hypothesis, I am not sure if any of the claims will hold up in tasks with real datasets. It may be helpful to have at least one such experiment where the data is not generated in the paper, and is based on an existing sequence-to-sequence modeling task.

**Questions:**

Please refer to the weaknesses above. It will be helpful to clarify them for the rebuttal.


- The paper is extremely difficult to read and understand because most of the sections do not include a broad high level context and directly dive into technical details with math/notations. Is it possible to provide relevant context in each sub-section? Also, is it possible to include a more detailed treatment of TA?

- The related works is minimal /missing. Several prior works that have attempted to understand neural network dynamics, and in particular those that have tried to understand the dynamics of recurrent models need to cited, and discussed. It will help to have a separate related works sections for this.

- The experiments are on very toy settings. While they are helpful in understanding each hypothesis, I am not sure if any of the claims will hold up in tasks with real datasets. It may be helpful to have at least one such experiment where the data is not generated in the paper, and is based on an existing sequence-to-sequence modeling task.

- Is there a reason why the paper considers RNNs alone, instead of (or in addition to) more common/widely used transformer models, for the analysis? Will there be significant differences in developing similar analyses for the latter?

---

### Official Review · Reviewer_cxSU · 2023-11-04

**Soundness:** 3 good
**Presentation:** 2 fair
**Contribution:** 2 fair
**Rating:** 3
**Confidence:** 4

**Summary:**

The paper explores how timing information is encoded in RNNs and how this knowledge gradually emerges during the training process. The authors conduct an in-depth analysis of the learning dynamics of two-state time-dependent automata in very simple RNNs. They also perform detailed analyses of the hidden states of these automata to uncover key aspects of the acquired solutions using PCA and an examination of the eigenvectors of the Jacobian of the hidden states. Although the paper predominantly concentrates on the acquisition of periodic time-dependent patterns, it also briefly touches on the consideration of relative time dependence.

**Strengths:**

The straightforward task and setup used in the paper enable the monitoring of relevant task inputs both during and after training. They also allow for the analysis of the RNN representations from a dynamical systems perspective.

**Weaknesses:**

- The task and settings are designed as a simplified framework for studying the influence of time, and some may argue that they are overly simplistic. While this setup allows for a comprehensive exploration of the system's behavior, I remain unconvinced that learning a two-state automaton (TA) truly represents a meaningful real-world sequence learning challenge. I am uncertain about the extent to which the observed results can be applied to real-world datasets.

- The paper concentrates on a particular RNN architecture and a specific activation function, potentially constraining the applicability of its findings to other architectures and tasks. Since learning dynamics are closely tied to the chosen RNN architecture, I believe the paper lacks comprehensiveness without addressing potential variations across RNN types. Numerous RNN architectures have been introduced that handle temporal information differently from a standard RNN (or GRU/LSTM).

- In Figure 4, it's unclear which parameters and their corresponding values led to the observed bifurcation. Given that the time-dependent system in this paper relies on inputs, these inputs can also influence the Jacobian matrix and subsequently impact the stability of the fixed points over time. Therefore, additional information is required to better understand the discussed bifurcation.

- Regarding the sentence (page 18) "These trajectories are quasi-periodic but approach period P as training progresses", it's not entirely clear how we can verify that the observed behaviors are genuinely quasi-periodic. They could also be higher-order periodic orbits converging towards a lower period P as training advances. One method to confirm this statement is to delve into the Lyapunov exponent spectra and the maximum Lyapunov exponent. Otherwise, how one can validate this assertion?

**Questions:**

Can the results of this paper be generalized to more complex settings?

---

### Meta-Review · Area_Chair_GmWh · 2023-12-10

**Metareview:**

Reviewers raised major concerns about the completeness of the study. The authors did not provide rebuttals. AC votes for rejection.

**Justification For Why Not Higher Score:**

The authors did not address the concerns raised by the reviewers.

**Justification For Why Not Lower Score:**

N/A

---

### Decision · Program_Chairs · 2024-01-16

Reject